# Depth Optimization Analysis of Articulated Steering Hinge Position Based on Genetic Algorithm

**Bing-wei Cao, Xin-hui Liu, Wei Chen \*, Yong Zhang and Ai-min Li**

School of Mechanical and Aerospace Engineering, Jilin University, Changchun 130022, China;
caobw16@mails.jlu.edu.cn (B.-w.C.); liuxh@jlu.edu.cn (X.-h.L.); zhangyong17@mails.jlu.edu.cn (Y.Z.);
liaimin5634@163.com (A.-m.L.)
\* Correspondence: chenwei_1979@jlu.edu.cn; Tel.:+86-186-4307-7790

**Abstract:** Articulated steering is affected by the position of the articulated points of the steering cylinder. When the two steering cylinders turn, there is a stroke difference and arm of force difference. The existence of the above differences causes the pressure fluctuation of the steering system. Firstly, the mathematical model of the steering mechanism is established through theoretical analysis. Then, the coordinates of the hinge points of the steering cylinder are optimized using genetic algorithm (GA) with the stroke difference function and cylinder pressure function as the objective functions. The curves of the stroke difference and the arm of force difference of the steering cylinder are obtained by mathematical modeling, and the correctness of the GA is verified. According to the optimization results, the wheel loader prototype was reconstructed, and the reconstruction verified by corresponding sensors. The experimental curves show that the steering system has no obvious pressure fluctuation. Finally, the arm of force difference and stroke difference curves were analyzed, and it was concluded that the arm of force difference was the main cause of pressure fluctuation. The objective function was improved, and the arm of force function and cylinder pressure function were taken as the objective functions to continue the optimization by GA. The arm of force difference and stroke difference after optimization were reduced, which provides a constructive reference for the design of articulated steering systems in the future.

**Keywords:** stroke difference; arm of force difference; improvement of objective function; GA

## 1. Introduction

A large number of steering experiments of wheel loaders show that there is always obvious pressure fluctuation in steering systems [1–3]. Due to the frequent steering operation of wheel loaders, the long-term existence of this phenomenon impacts hydraulic components, produces vibration and noise, and seriously affects the stability and reliability of steering systems [4–6].

Zhang [7] established the mathematical model of steering systems by using equivalent parameters. Starting from the frequency characteristics of the system and the stability of the system, the pressure oscillation phenomenon of the steering hydraulic system was analyzed, and the causes and influencing factors of the pressure fluctuation during steering were determined. Zhu [8], in a study based on the LG953 wheel loader, used the SQP algorithm to optimize the hinge position of steering mechanisms, and obtained the corresponding stroke difference and arm of force difference curves. However, in the SQP algorithm, it is easy to enter the local optimal solution, and it has fewer added constraints, which affects the selection of the optimal solution. The objective function of Gui [9] was to minimize the stroke difference and the power of the oil pump. The steering mechanism was optimized by using the mixed penalty function method, so that there was no stroke difference in the optimized steering mechanism. However, the steering resistance distance was derived from empirical formulas, and the

influence of steering angle on the steering resistance distance was not considered [10,11]. The above analysis did not consider the influence of the stroke difference and arm of force difference caused by the position of the hinge point of the steering mechanism on the pressure fluctuation of the steering system. In this paper, the reasonable position of the hinge point of the steering mechanism is also found, the relationship between the stroke difference and the arm of force difference is analyzed, and the main influencing factors of the pressure fluctuation of the steering system are found.

On the premise of satisfying the space limitation and driving moment, the coordinates of many groups of points can be obtained. There must be a set of coordinate points in these groups of point coordinates that minimizes the stroke difference and arm of force difference of the steering mechanism and finds the optimal solution among many solutions. The genetic algorithm (GA) is a useful algorithm for reserving useless or removing simulated biological evolution in the process of optimization [12]. It can be used to search for the optimal articulation coordinates [13–15]. The GA starts from the collection of clusters and has a large coverage, which is conducive to global optimization. It can deal with multiple individuals in a group at the same time and reduce the risk of falling into the local optimal solution. The GA can optimize the solution of the problem generation after generation, and approximate the optimal solution. Sivaram [16] aimed to solve the vehicle routing problem with time-limited windows by using GA to minimize the total distance and the total number of vehicles. The optimization results of the GA were verified by experiments. Johannes Knust [17] finished the preform optimization for hot processes using GA, and his study demonstrated that a GA-based approach could realize the optimization effectively. Duan [18], aiming at the job shop scheduling problem and taking the maximum lead time and the minimum tardiness as the optimization objectives, obtained the solution method of the GA and verified the effectiveness of the algorithm by simulation.

Optimal results can be obtained by only using the stroke difference function and arm of force difference function as objective functions; however, the steering system pressure will increase and energy will be wasted. Therefore, the selection of the objective function was the key to getting the optimal solution of the GA in this paper. The calculation formula of the steering resistance distance was obtained through theoretical analysis. In order to obtain the optimal coordinates of articulation points, the stroke difference function and cylinder pressure function were selected as the objective functions, and the arm of force difference function as the constraint condition function when GA was used the first time. The purpose of this method was to minimize the system pressure while obtaining the optimal stroke difference and arm of force difference. Then, the arm of force difference curve and stroke difference curve were analyzed. It was found that the arm of force difference was the main factor affecting the pressure fluctuation. Then, the objective function of the GA was reformed again. The optimization results showed that the arm of force difference and stroke difference were lower than in the first optimization, which provides a reference for the design of the steering mechanisms of wheel loaders in the future.

## 2. Mechanism Analysis of Stroke Difference and Arm of Force Difference

In the steering process of wheel loaders, the relation between the front and rear frame around the hinge joint is realized by the elongation of the outer cylinder and the synchronous shortening of the inner cylinder. A schematic diagram of the steering mechanism is shown in Figure 1. Points A and B are the hinge points of the steering cylinder with the front frame, points C and D are the hinge points of the steering cylinder with the rear frame, and point O is the hinge point of the front and rear frames. For simplification, we represent $\frac{\overline{AB}}{2}$ as $x_f$ and $\frac{\overline{CD}}{2}$ as $x_r$. The distance from the center of the connection between the two hinge points of the front frame to the hinge point O is represented as $y_f$, and the distance between the center of the connection of the two hinge points of the rear frame to the hinge point O of the rear frame is represented as $y_r$.

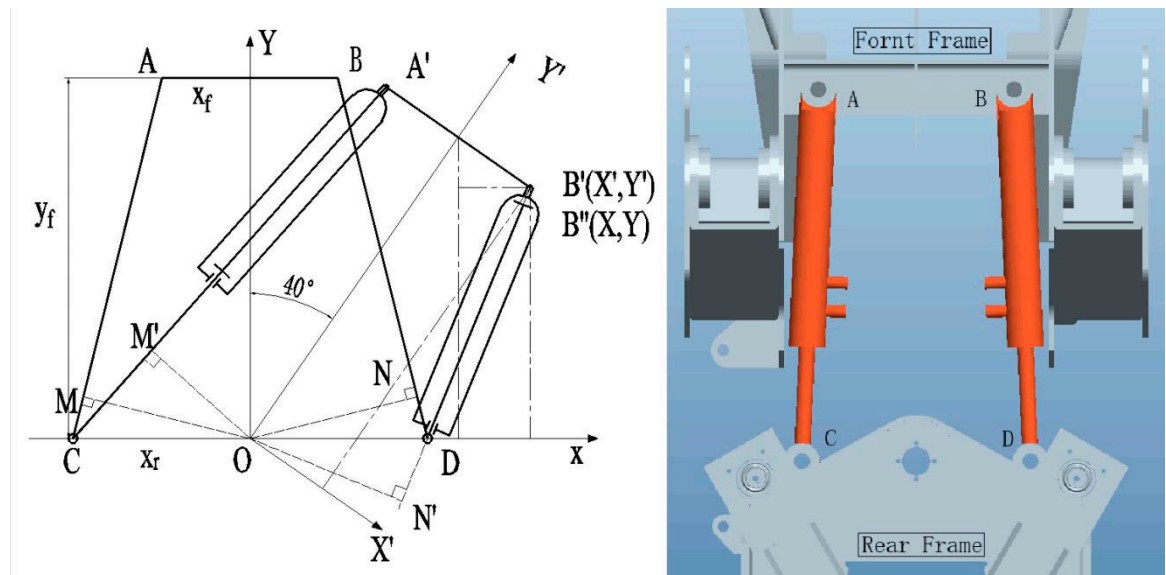

**Figure 1.** Schematic diagram of the steering mechanism.

With the change of the steering angle, the extension of AC on the left cylinder and the shortening of BD on the right cylinder are shown in Formulas (1) and (2), respectively. The difference between them is the stroke difference.

$$\Delta L_l = \overline{A''C} - \overline{AC} = \sqrt{(-x_f \cos\theta + y_f \sin\theta + x_r)^2 + (x_f \sin\theta + y_f \cos\theta - y_r)^2} \\ - \sqrt{(x_r - x_f)^2 + (y_r + y_f)^2} \tag{1}$$

$$\Delta L_r = \overline{B''D} - \overline{BD} = \sqrt{(x_f \cos\theta + y_f \sin\theta - x_r)^2 + (y_f \cos\theta + x_f \sin\theta - y_r)^2} \\ - \sqrt{(x_r - x_f)^2 + (y_r + y_f)^2} \tag{2}$$

According to Helen's formula [19], the force arms $\overline{OM\prime}$ and $\overline{ON\prime}$ of the two cylinders are obtained respectively, and the difference between them is the arm of force difference.

$$\begin{aligned} \overline{OM'} &= \frac{2\sqrt{P(P-\overline{OA'})(P-\overline{OC})(P-\overline{A'C})}}{\overline{A'C}} \\ \overline{ON'} &= \frac{2\sqrt{S(S-\overline{OB'})(S-\overline{OD})(S-\overline{B'D})}}{\overline{B'D}} \end{aligned} \tag{3}$$

where:

$$\begin{aligned} P &= \frac{\overline{OC}+\overline{OA'}+\overline{A'C}}{2}, S = \frac{\overline{OD}+\overline{OB'}+\overline{B'D}}{2}, \overline{OA'} = \overline{OA}, \overline{OB'} = \overline{OB}, \\ \overline{A'C} &= \overline{AC} + \Delta L_l, \overline{B'D} = \overline{BD} + \Delta L_r \end{aligned} \tag{4}$$

The calculation of steering resistance distance is generally based on the empirical formula, but the empirical formula has its limitations [19]. In this paper, the steering resistance distance was calculated. The analysis showed that the steering resistance distance consisted of three parts: torque when the tire pivots around the center of its contact patch $M_m$, the resistant torque caused by the opposite steering of the left and right wheels $M_g$, and the torque generated by the tangential force on the rear axle $F_r$. As shown in Figure 2, the formulas for calculating the steering resistance moment based on the virtual displacement principle are as follows:

$$\begin{cases} \frac{L_1}{\sin\beta} = \frac{L_1+L_2}{\sin\theta} \\ \theta = \alpha + \beta \end{cases} \Rightarrow \begin{cases} \partial\alpha = (1 - \frac{k\cos\theta}{\sqrt{1-k^2\sin^2\theta}}) \cdot \partial\theta \\ \partial\beta = \frac{k\cos\theta}{\sqrt{1-k^2\sin^2\theta}} \cdot \partial\theta \end{cases} \tag{5}$$

$$T \cdot \partial\theta - M_F \cdot \partial\alpha - M_R \cdot \partial\beta - F_r \cdot \partial r = 0 \tag{6}$$

where $T$ is the steering torque, $M_F$, $M_R$ are the front and rear resistance torques, $k = \frac{L_1}{L_1+L_2} = 0.5$.

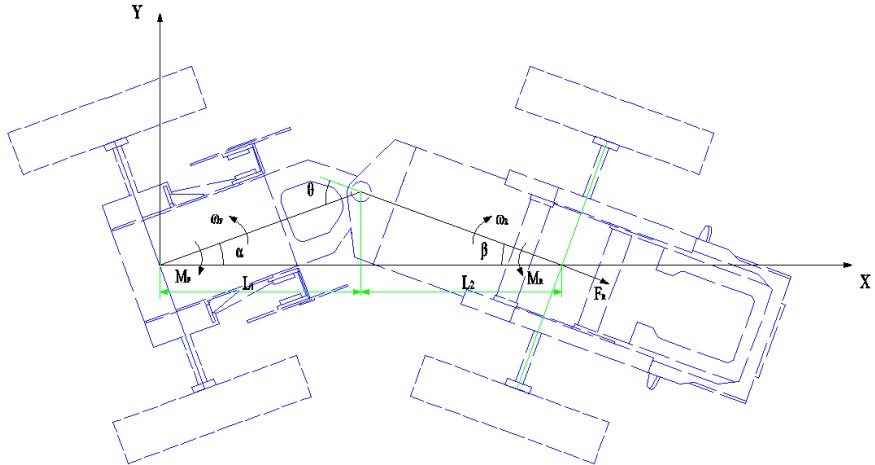

**Figure 2.** Calculation of steering resistance distance.

Through coordinate system transformation, $A''$ coordinates of $A'$ in the coordinate system XOY and $B''$ coordinates of $B'$ in the coordinate system XOY were obtained. Then, the mathematical model of the steering mechanism was established by the Simulink module in MATLAB, as shown in Figure 3. The purpose of this mathematical model is to verify the position of articulation points optimized by GA, and to obtain the curve of the stroke difference of the steering cylinder with the steering angle, the curve of arm of force difference with the steering angle, the curve of the resistance moment of the steering system with the steering angle, and the curve of the pressure of the steering system with the steering angle. The model can also be used to calculate the type selection of the steering system to determine the size of the steering cylinder and the pressure of the steering hydraulic system, and provide help for the design of the steering hydraulic system.

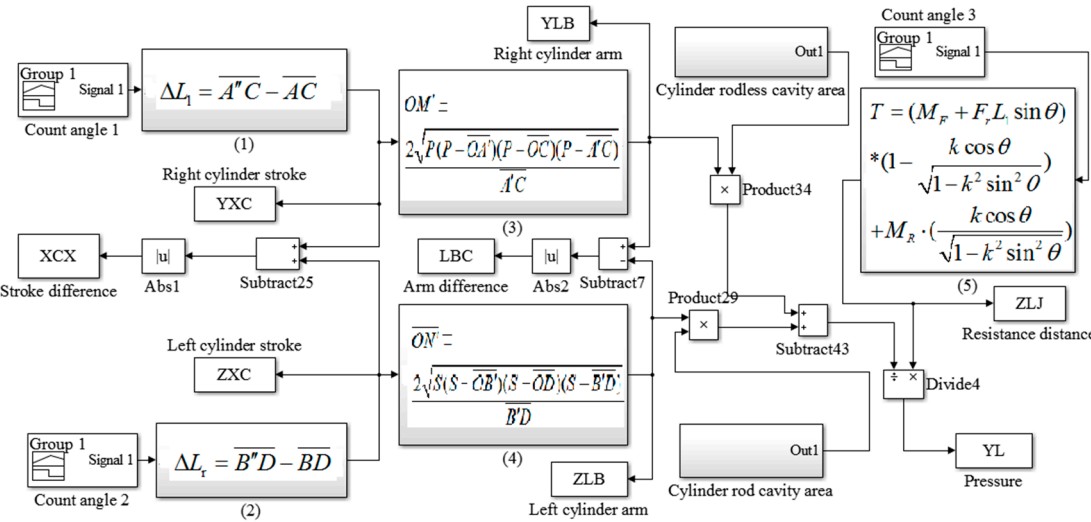

**Figure 3.** Model of the steering mechanism.

## 3. Optimal Selection of Hinge Point Coordinates

From the above analysis, it can be seen that the reasonable coordinate position of articulation points can put the stroke difference and arm of force difference within a reasonable range, thus reducing the pressure fluctuation of the steering system, to find the optimal solution that satisfies the conditions. GA is a search heuristic algorithm which can be used to solve the optimization and is often used to

search for the problem and find the optimal solution [20]. This paper uses GA to optimize the selection of the hinge position of the steering mechanism. The optimization process of the GA combined with this example is shown in Figure 4 below.

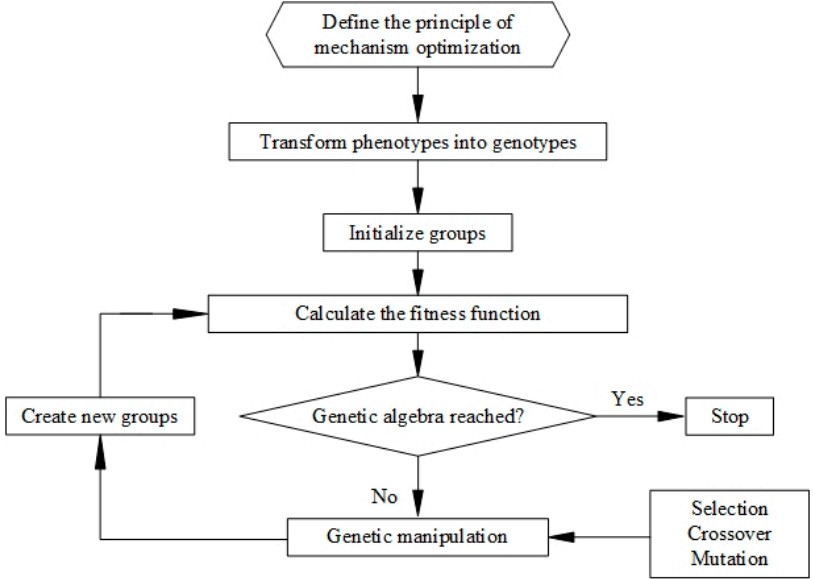

**Figure 4.** The genetic algorithm optimizing solution steps.

Following is an introduction to the main steps of the GA according to Figure 4.

Step 1: Clarify the optimal design principle of the steering mechanism. The number of the primary population is 1000, the crossover probability is 0.5, and the mutation probability is 0.02.

Step 2: Binary coding and chromosome length calculation, that is, phenotype into genotype. The numbers in parentheses are the boundary constraints of the steering mechanism.

$$
\begin{cases}
m1 = RequiredStringlength(150, 350) \\
m2 = RequiredStringlength(930, 1160) \\
m3 = RequiredStringlength(300, 350) \\
m4 = RequiredStringlength(0, 30)
\end{cases}
\tag{7}
$$

$$
m = m1 + m2 + m3 + m4
\tag{8}
$$

Step 3: Initialize the population and construct the fitness function. Since the selection of the fitness function is the key to getting the optimal solution of the GA, the construction of the fitness function is emphasized here. The construction of the fitness function is based on the combination of the objective function and the constraint function.

(1) The objective function in this example is

$$
Objvaue = W_1 \cdot f_1(x) + W_2 \cdot f_2(x)
\tag{9}
$$

In the formula, $W_1$ is the weighting factor of the minimum score object function of the steering cylinder stroke difference; $W_2$ is the weighting factor of the minimum sub-objective function of the steering cylinder pressure function; $f_1(x)$ is the objective function of minimizing the stroke difference of the steering cylinder; and $f_2(x)$ is the cylinder pressure function.

In this paper, the cylinder pressure function was taken as one of the objective functions because the curve of the steering pressure with the steering angle can be calculated from the steering resistance distance and the cylinder arm of force. As the objective function, the maximum arm of force value can

be obtained on the premise of overcoming the steering resistance distance, thus reducing the steering hydraulic system pressure and consequently saving on energy consumption [21,22].

(2) The steering system can operate normally without dead angle and interference. The constraints to be met are as follows [23,24]:

(a)　The boundary constraints of each hinge point are shown in Formula (6).

(b)　The constraint of the arm of force difference is $(\overline{OM'} - \overline{OM} - 15; \overline{ON'} - \overline{ON} - 15)$.

(c)　The transmission angle of the two steering cylinders is between $10° \sim 170°$.

(d)　The stability constraints of the steering cylinder allows the expansion ratio of the steering cylinder to satisfy the following formula:

$$1.3 \leq \frac{L_{\max}}{L_{\min}} \leq 1.6 \tag{10}$$

The constraint function constructed by the above constraints is as follows:

$$C(x) = \sum_{i=1}^{k} power(zuida(c(1),0),2) \tag{11}$$

where $k$ is the number of constraint functions, and the fitness function built by the constraint condition is added by the penalty function method.

Combining the above constraint function with the objective function and taking the opposite approach, the fitness function is finally constructed as follows:

$$F(x)\frac{1}{Objvaue + C(x)} \tag{12}$$

Step 4: Calculate the fitness function, calculate the possibility of individuals entering the next generation by roulette, and then simulate crossover and mutation. Crossover and mutation can be obtained by writing corresponding functions in MATLAB.

Step 5: Generate a new generation and continue to calculate the fitness function until the optimal result is obtained.

The curve of the fitness function is shown in Figure 5. The maximum value of the fitness function was 0.35; the maximum emerged in the 43rd generation. The position coordinates of each hinge point optimized by GA are shown in Table 1. Table 1 also contains a comparison with the original data.

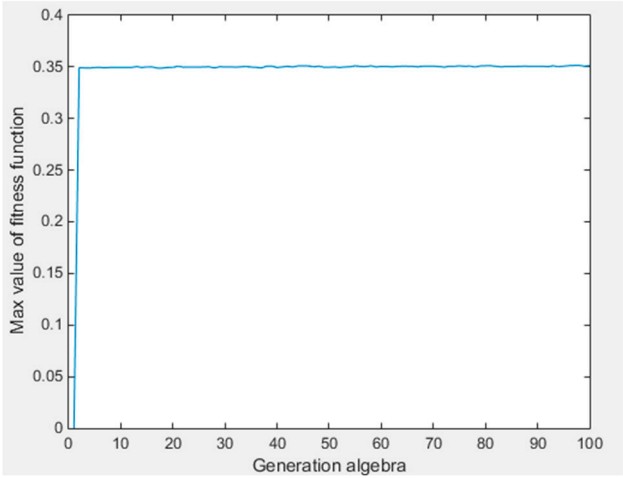

**Figure 5.** The fitness function.

**Table 1.** The coordinates of articulated points optimized by GA.

| Parameter | After Optimization | Original |
|---|---|---|
| 274 mm | 150 mm | |
| 1030 mm | 1160 mm | |
| 320 mm | 360 mm | |
| 0 mm | 0 mm | |
| Maximum stroke difference | 2.1 mm | 26 mm |
| Maximum arm of force difference | 12 mm | 56 mm |
| Cylinder parameter | 80/45/411 mm | 90/50/485 mm |

Aiming at the stochastic problem of the genetic algorithm, 10 sets of optimal values were calculated, and the average and standard deviation (SD) were calculated, as shown in Table 2.

**Table 2.** The average and standard deviation.

| 1 | 2 | 3 | 4 | 5 | 6 | 7 | 8 | 9 | 10 | Average | SD |
|---|---|---|---|---|---|---|---|---|---|---|---|
| 275 | 273 | 274 | 278 | 276 | 274 | 275 | 274 | 273 | 274 | 274 | 1.4 |
| 1030 | 1031 | 1029 | 1030 | 1027 | 1029 | 1030 | 1032 | 1031 | 1030 | 1030 | 1.3 |
| 322 | 320 | 325 | 323 | 320 | 321 | 320 | 320 | 323 | 321 | 321 | 1.6 |
| 0.6 | 0.3 | 0 | 0.1 | 0.2 | 0 | 0.8 | 0.6 | 0.7 | 0 | 0.3 | 0.3 |
| 0.3515 | 0.3514 | 0.3513 | 0.3512 | 0.3515 | 0.3513 | 0.3515 | 0.3513 | 0.3512 | 0.3511 | 0.3513 | 0 |

As can be seen from Table 2, when the fitness function value tends to be stable, the optimized coordinate points tend to be stable.

The position of articulation points in Table 1 is brought into the mathematical model of the steering mechanism in Figure 3. The curve of stroke difference of the steering cylinder versus the steering angle is shown in Figure 6, and the curve of the arm of force difference of the steering cylinder versus the steering angle is shown in Figure 7.

The force arm and the stroke of the steering cylinder varied with the steering angle, as shown in Figures 8 and 9, respectively. The stroke of the steering cylinder was 411 mm, as calculated from Figure 8. The cylinder diameter of the steering cylinder was 80 mm and the rod diameter was 45 mm, as calculated from Figure 9 and Formula (5). In order to get enough driving moment, the cylinder size of the original steering system was 90/50 mm. In this paper, by establishing the cylinder pressure function, we can get the optimal arm of force difference and maximize the arm value. The size of the cylinder further verifies the accuracy of the objective function in GA.

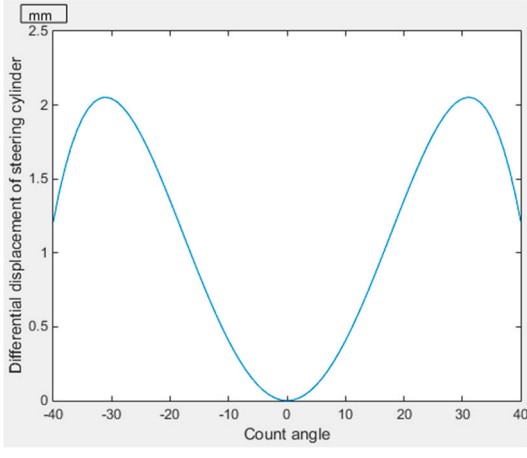

**Figure 6.** Curve of the stroke difference of a cylinder with steering angle.

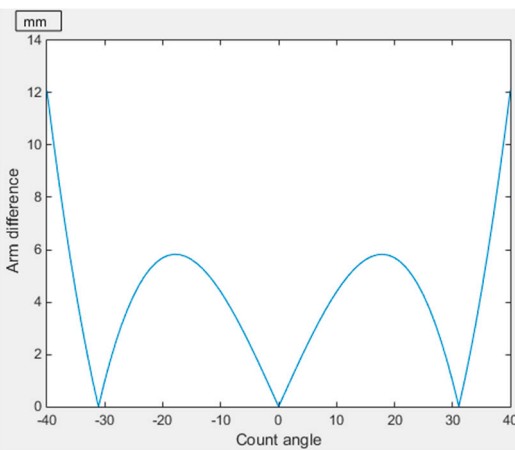

**Figure 7.** Curve of the arm of force difference of a cylinder with steering angle.

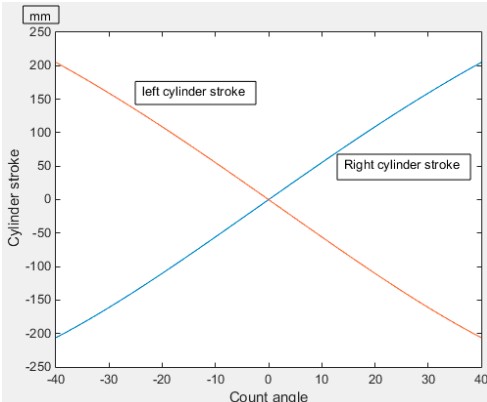

**Figure 8.** Variation curve of the steering cylinder stroke with steering angle.

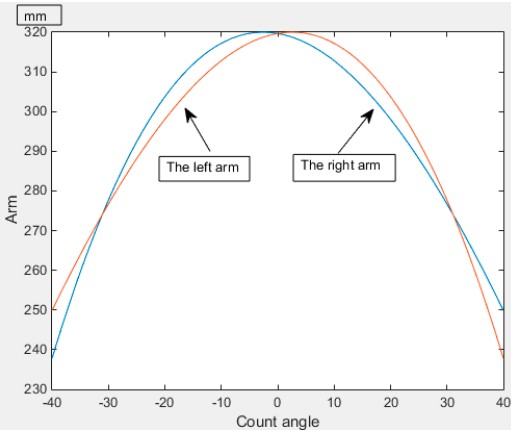

**Figure 9.** Variation curve of the steering cylinder arm with steering angle.

## 4. Real Vehicle Modification and Test Verification

In the above analysis, the results of GA optimization were verified by mathematical modeling, but lacked experimental support. In this paper, through the school–enterprise cooperation project, the articulation position of the wheel loader prototype car obtained by GA was reconstructed, as shown in Figure 10. Next, the stroke difference of the optimized steering system was validated by connecting displacement sensors, and the arm of force difference of the optimized steering system was validated by connecting pressure sensors. After connecting the above sensors, the wheel loader prototype was

started and the in situ steering test was carried out. The experimental steering cylinder stroke curve and pressure curve are shown in Figure 11.

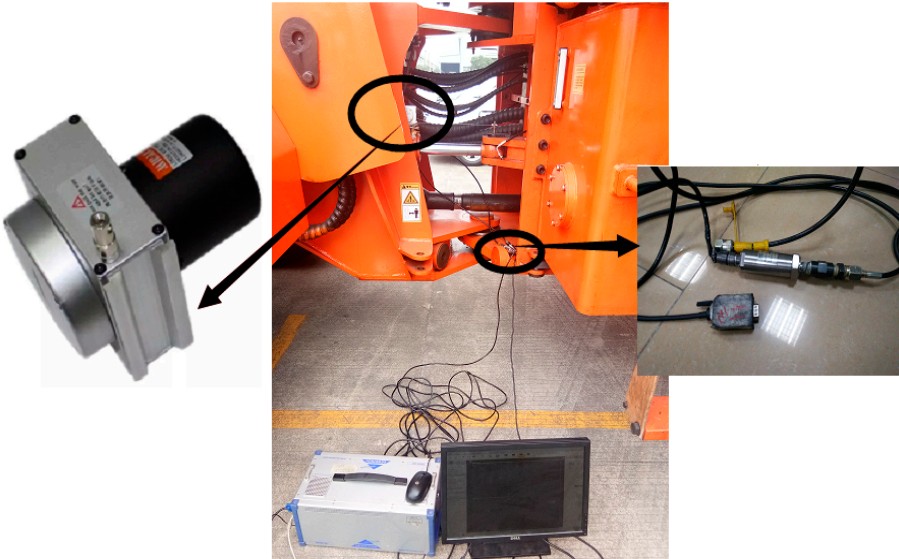

**Figure 10.** Wheel loader prototype test.

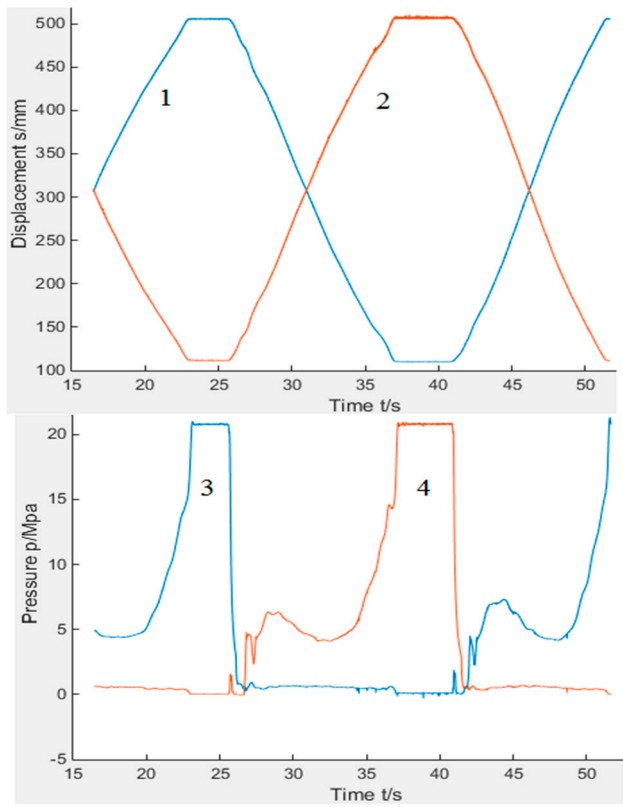

**Figure 11.** Vehicle test curve. (**1**) Left steering cylinder displacement; (**2**) right steering cylinder displacement; (**3**) left steering cylinder, no rod cavity pressure; (**4**) right steering cylinder, no rod cavity pressure.

From the experimental curve of the wheel loader prototype shown in Figure 11, it can be seen that (1) the steering system pressure was relatively stable, the average pressure was about 5 MPa, and there was no sharp fluctuation; and (2) the pressure of the unilateral steering cylinder increased first and

then decreased, which corresponds to the arm of force curve of the steering mechanism. The above analysis shows the rationality of the position of articulation points optimized by GA.

## 5. Deep Optimization Analysis

Above all, the rationality of the coordinates of the articulated points optimized by GA were verified by mathematical modeling and the wheel loader prototype experiment. Next, the stroke difference and arm of force difference of the steering mechanism were measured, and the influence of the two on the pressure fluctuation of the steering system was further explored. As can be seen from Figures 6 and 7, when the arm of force difference of the steering mechanism is zero (the corresponding steering angle is 32 degrees), the stroke difference of the steering cylinder reaches the maximum value of 2 mm. According to the experimental curve of the wheel loader prototype shown in Figure 11, the pressure fluctuation of the steering cylinder also exists at 27 s and 43 s. Therefore, it is necessary to analyze the above phenomena in order to further optimize the steering mechanism.

When steering, as shown in Figure 12, the steering cylinder has both a telescopic motion and a swing around the front and rear articulation points.

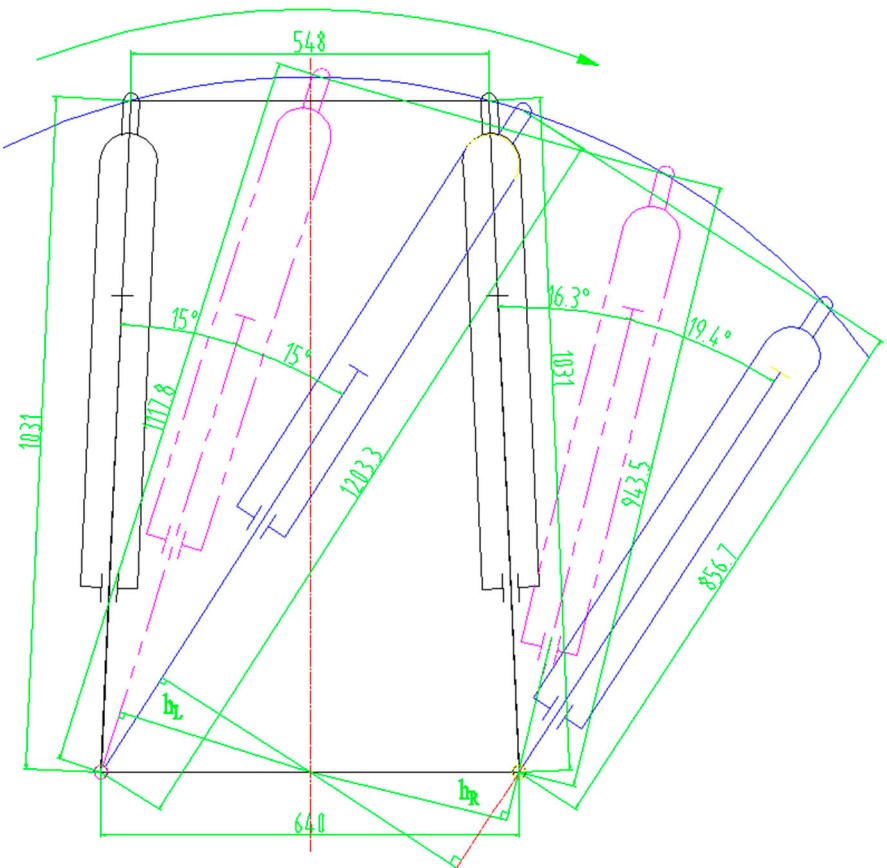

**Figure 12.** Steering mechanism.

When turning angle $\theta$, and the stroke of the left steering cylinder and the arm of force of the left cylinder are $L_L$ and $h_L$, respectively:

$$L_L = \sqrt{x^2 + R^2 - 2 \cdot x_r \cdot R \cdot \cos(\alpha + \theta)} \tag{13}$$

$$h_L = \frac{R \cdot x_r \cdot \sin(\alpha + \theta)}{L_L} \tag{14}$$

After the derivation of Formula (12), Formula (13) is introduced to obtain

$$v_L = h_L \cdot \omega$$
$$v_R = h_R \cdot \omega$$

(15)

In the formula, $v_L$ and $v_R$ are the left and the right steering cylinder speeds, respectively, and $\omega$ is the steering angular speed of the front frame.

Formula (14) shows that the speed of the two cylinders is the same when the arms of force of the left and right cylinders are the same but, as shown in Figure 12, the rotation angles of the two cylinders relative to their own articulation points are different, which results in the greatest stroke difference when the arms of force of the steering mechanism are the same. Therefore, the arm of force difference of the steering mechanism should be optimized to the greatest extent. The objective functions of the arm of force difference and the cylinder pressure function are constructed in the GA, while the other constraint objective functions remain unchanged. The curves of the distance difference of the steering mechanism with the steering angle and the arm of force difference with the steering angle are shown in Figures 13 and 14.

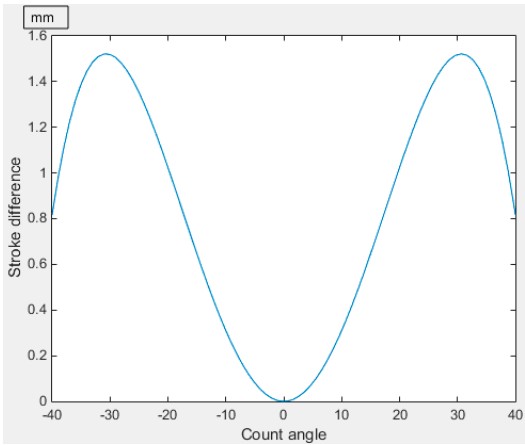

**Figure 13.** Curve of the stroke difference of a cylinder with steering angle.

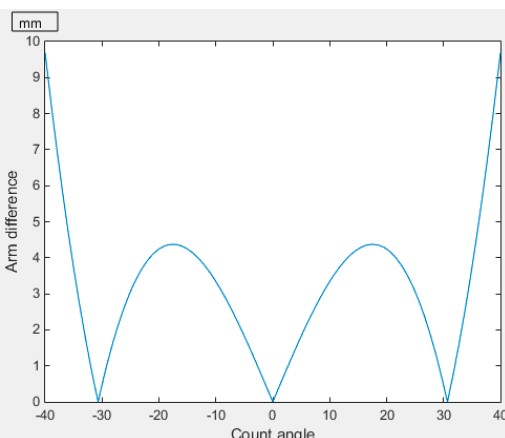

**Figure 14.** Curve of the arm of force difference of a cylinder with steering angle.

From Figures 13 and 14, it can be seen that the maximum stroke difference after optimization is 1.5 mm, and the corresponding stroke difference is 0.8 mm when the steering angle is 40°. The maximum arm of force difference is 9.8 mm and the corresponding arm of force difference is 4.2 mm when the steering angle is 19°. Compared with Figures 6 and 7, the maximum stroke difference is reduced by 0.5 mm and the corresponding stroke difference is reduced by 0.5 mm when the steering

angle is 40°; the maximum force arm difference is reduced by 2.2 mm and the corresponding force arm difference is reduced by 2.2 mm when the steering angle is 19°. The above results show that further optimization of the arm of force difference can reduce the stroke difference, which verifies the correctness of the theoretical analysis. When turning to the limit position, the pressure fluctuation is easily to produced due to the effect of the steering limit, the arm of force difference and stroke difference are reduced when turning to the limit position compared with the previous optimization results, and the influence of the pressure fluctuation is further reduced. Subsequently, the prototype should be modified according to the results of the second optimization, and the correctness of the above conclusions should be verified by experimental data.

## 6. Conclusions

(1) To find the causes of pressure fluctuation in steering systems, a mathematical model of steering mechanism was established, analyzing the stroke difference and the arm of force difference. Using the stroke difference and the pressure function as objective functions, the position of the articulation points of the steering mechanism was optimized by GA. The curves of the stroke difference, arm of force difference, and stroke and arm change with steering angle were obtained. The wheel loader prototype was reconstructed according to the optimized hinge position of the steering cylinder. After connecting the displacement and pressure sensors, the correctness of the mathematical model and the genetic algorithm were verified.

(2) By analyzing the stroke difference, the arm of force difference, and the vehicle test curve, it was found that the stroke difference was the largest when the arm of force difference was the smallest. This phenomenon was analyzed theoretically. The analysis results show that the arm of force difference was the main factor causing pressure fluctuation. The objective function of the GA was reformed. The new objective function is composed of the arm of force difference and the cylinder function. The position of articulation points and the corresponding arm of force difference and stroke difference curves of the steering mechanism were obtained. The maximum stroke difference was reduced by 0.5 mm and the maximum arm of force difference was reduced by 2.2 mm.

**Author Contributions:** Conceptualization, methodology, B.C., X.H. and W.C.; software, B.C. and Y.Z.; validation, B.C., Y.Z. and A.L.; formal analysis A.L.; investigation, resources, B.C. and W.C.; data curation, B.C. and Y.Z.; writing—original draft preparation, B.C.; writing—review and editing, X.H. and W.C.; visualization, Y.Z. and A.L.; supervision, W.C.; project administration, B.C., X.H. and W.C.; funding acquisition, W.C.

**Funding:** This research was funded by The National Key Research and Development Program of China under Grant No. 2016YFC0802904.

**Conflicts of Interest:** The authors declare no conflict of interest.

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
