# Peer review of "Depth Optimization Analysis of Articulated Steering Hinge Position Based on Genetic Algorithm"

_algorithms, doi:10.3390/a12030055_

Round 1

Reviewer 1 Report

The problem solved by the authors is interesting and practically relevant.  It should be clearly introduced however earlier in the paper.  For example, a master figure showing the articulated vehicle with the hydraulic cylinders in place (i.e. a combination of Fig.2 and Fig.1) is missing.

Id data exists, a comparison of the experimental results for the initial and the improved articulated steering would be a good addition to the paper.  

Some MATLAB plots (Fig. 8,9,13,14) do not show units on the Y axes.  

Also consider including in the reference list of the following references:  

https://www.sciencedirect.com/science/article/pii/0022489889900256

https://www.sciencedirect.com/science/article/pii/S2288430017300519

https://journals.sagepub.com/doi/abs/10.1177/0954407017729052?journalCode=pidb

https://www.tandfonline.com/doi/abs/10.1080/00423114.2011.622904

http://mechanicaldesign.asmedigitalcollection.asme.org/article.aspx?articleid=1450786

Author Response

The questions you raised are very meaningful. The attachment is a revision of the questions I raised for you.

Reviewer 2 Report

Present first the object of study. The reader needs to understand what it is intended to be optimized. Present the initial design or similar constructions.

Improve the bibliography, especially for the estimation of the necessary steering torque.

Present the most difficult tractor working conditions (for the steering).

Improve the figures, using larger and clearer notations. Indicate all the necessary elements in the figures. Indicate measurement units.

Introduce the used terms (stroke difference, arm of force, steering resistance distance, etc), explaining and schematizing them.

Explain how the used equations and the functions were obtained. Indicate haw were chosen some weights (the objective function) and input data (in the use of the genetic algorithm).

Make plots and kinematic schemes presenting in the same place the differences between the initial and optimized designs. Present how the design change from different generations.

I appreciate a lot the entire work, but the presentation is incomplete and twisted (unnatural) and so very difficult to follow.

Good luck!

Author Response

The questions you raised are very meaningful. The attachment is a revision of the questions I raised for you.

Thank you!

Round 2

Reviewer 1 Report

It appears that you used "count angle", "steering angle" or just "angle" interchangeably throughout the paper.  Please use only one i.e. steering angle.  

When describing plots, you may also want to use the formulations with "as function of" or using "versus" (abbreviated "vs.").  See for example:

https://d2vlcm61l7u1fs.cloudfront.net/media%2F7ab%2F7ab52902-df28-4a10-83ef-48fed2a8c3ae%2Fphp2VBrZl.png

http://www.batesville.k12.in.us/physics/APPhyNet/Measurement/Images/f_vs_a_graph.gif

https://d2vlcm61l7u1fs.cloudfront.net/media%2Fc26%2Fc262681b-d381-432c-9055-b0c9daf4c856%2Fphpc3JQ4I.png

Author Response

Thank you for your meticulous work. According to your suggestions, the article has been revised, as shown in the annex.

Thank you.

Reviewer 2 Report

Now is much better. Thank you for responding to my suggestions.

1. An overall view of the loader (upper view) is still missing in the introductory part of the article.

2. Indicate a reference for the "Helen's formula" (line 96).

3. Please consider to replace lines 104-105 with:
"torque when the tire pivots around the center of its contact patch Mm, the resistant torque caused by the opposite steering of the left and right wheels Mg, and the torque generated by the tangential force on the rear axle Fr.

4. It is still not indicated how the Weighting Factors W1 and W2 (line 149, equation 8) are adopted. These values are extremely important for the optimization results.

Congratulations for yours good work!

Author Response

Thank you for your meticulous work. According to your suggestions, the article has been revised, as shown in the annex.

Thank you!

Round 3

Reviewer 1 Report

The MATLAB produced plots can be edited using PAINT or other raster graphics editor and change "Count angle"  to "Steering angle" and "Generation algebra"  to "Generation" 

Reviewer 2 Report

Congratulations!